# Vocabulary Selection Strategies for Neural Machine Translation

**Gurvan L'Hostis**[*]
École polytechnique
Palaiseau, France

**David Grangier**
Facebook AI Research
Menlo Park, CA

**Michael Auli**
Facebook AI Research
Menlo Park, CA

## Abstract

Classical translation models constrain the space of possible outputs by selecting a subset of translation rules based on the input sentence. Recent work on improving the efficiency of neural translation models adopted a similar strategy by restricting the output vocabulary to a subset of likely candidates given the source. In this paper we experiment with context and embedding-based selection methods and extend previous work by examining speed and accuracy trade-offs in more detail. We show that decoding time on CPUs can be reduced by up to 90% and training time by 25% on the WMT15 English-German and WMT16 English-Romanian tasks at the same or only negligible change in accuracy. This brings the time to decode with a state of the art neural translation system to just over 140 words per seconds on a single CPU core for English-German.

## 1 Introduction

Neural Machine Translation (NMT) has made great progress in recent years and improved the state of the art on several benchmarks (Jean et al., 2015b; Sennrich et al., 2016b; Zhou et al., 2016). However, neural systems are typically less efficient than traditional phrase-based translation models (PBMT; Koehn et al. 2003), both at training and decoding time.

The efficiency of neural models depends on the size of the target vocabulary and previous work has shown that vocabularies of well over 50k word types are necessary to achieve good accuracy (Jean et al., 2015a; Zhou et al., 2016). Neural translation systems compute the probability of the next target word given both the previously generated target words as well as the source sentence. Estimating this conditional distribution is linear in the size of the target vocabulary which can be very large for many language pairs (Grave et al., 2016). Recent work in neural translation has adopted sampling techniques from language modeling which do not leverage the input sentence (Mikolov et al., 2011; Jean et al., 2015a; Chen et al., 2016; Zhou et al., 2016).

On the other hand, classical translation models generate outputs in an efficient two-step *selection* procedure: first, a subset of promising translation rules is chosen by matching rules to the source sentence, and by pruning them based on local scores such as translation probabilities. Second, translation hypotheses are generated that incorporate non-local scores such as language model probabilities. Recently, Mi et al. (2016) proposed a similar strategy for neural translation: a selection method restricts the target vocabulary to a small subset, specific to the input sentence. The subset is then scored by the neural model. Their results demonstrate that vocabulary subsets that are only about 1% of the original size result in very little to no degradation in accuracy.

This paper complements their study by experimenting with additional selection techniques and by analyzing speed and accuracy in more detail. Similar to Mi et al. (2016), we consider selecting target words based either on a dictionary built from Viterbi word alignments, or by matching phrases in a traditional phrase-table, or by using the $k$ most frequent words in the target language. In addition, we investigate methods that do not rely on a traditional phrase-based translation model or alignment model to select target words. We investigate bilingual co-occurrence counts, bilingual embeddings as well as a discriminative classifier to leverage context information via features extracted from the entire source sentence (§2).

---

[*]Gurvan was interning at Facebook for this work.

Our experiments show speed-ups in CPU decoding by up to a factor of 10 at very little degradation in accuracy. Training speed on GPUs can be increased by a factor of 1.33. We find that word alignments as the sole selection method is sufficient to obtain good accuracy. This is in contrast to Mi et al. (2016) who used a combination of the 2,000 most frequent words, word alignments as well as phrase-pairs. Selection methods often fall short in retrieving all words of the gold standard human translation. However, we find that with a reduced vocabulary of $\sim 600$ words they can recover over 99% of the words that are actually chosen by a model that decodes over the entire vocabulary. Finally, the speed-ups obtained by vocabulary selection become even more significant if faster encoder models are used, since selection removes the burden of scoring large vocabularies (§4).

## 2 VOCABULARY SELECTION STRATEGIES

This section presents different selection strategies inspired by phrase-based translation. We improve on a simple word co-occurrence method by estimating bilingual word embeddings with Hellinger PCA and then by using word alignments instead of co-occurrence counts. Finally, we leverage richer context in the source via bilingual phrases from a phrase-table or by using the entire sentence in a Support Vector Machine classifier.

### 2.1 WORD CO-OCCURRENCES

This is the simplest approach we consider. We estimate a co-occurrence table which counts how many times each source word $s$ co-occurs with each target word $t$ in the training bitext. The table allows us to estimate the joint distribution $P(s,t)$. Next, we create a list of the $k$ target words that co-occur most with each source word, i.e., the words $t$ which maximize $P(s,t)$ for a given $s$. Vocabulary selection then simply computes the union of the target word lists associated with each source word in the input.

We were concerned that this strategy over-selects frequent target words which have higher co-occurrence counts than rare words, regardless of the source word. Therefore, we experimented with selecting target words maximizing point-wise mutual information (PMI) instead, i.e.,

$$PMI(s,t) = \frac{P(s,t)}{P(s)P(t)}$$

However, this estimate was deemed too unreliable for low $P(t)$ in preliminary experiments and did not perform better than just $P(s,t)$.

### 2.2 BILINGUAL EMBEDDINGS

We build bilingual embeddings by applying Hellinger Principal Component Analysis (PCA) to the bilingual co-occurrence count matrix $M_{i,j} = P(t = i | s = j)$; this extends the work on monolingual embeddings of Lebret & Collobert (2014) to the bilingual case. The resulting low rank estimate of the matrix can be more robust for rare counts. Hellinger PCA has been shown to produce embeddings which perform similarly to word2vec but at higher speed (Mikolov et al., 2013; Gouws et al., 2015).

For selection, the estimated co-occurrence can be used instead of the raw counts as described in the above section. This strategy is equivalent to using the low rank representation of each source word (source embedding, i.e., column vectors from the PCA) and finding the target word with the closest low rank representation (target embeddings, i.e., row vectors from the PCA).

### 2.3 WORD ALIGNMENTS

This strategy uses word alignments learned from a bilingual corpus (Brown et al., 1993). Word alignment introduces latent variables to model $P(t|s)$, the probability of source word $t$ given target word $s$. Latent variables indicate the source position corresponding to each target position in a sentence pair (Koehn, 2010). We use FastAlign, a popular reparameterization of IBM Model 2 (Dyer et al., 2013). For each source word $s$, we build a list of the top $k$ target words maximizing $P(t|s)$. The candidate target vocabulary is the union of the lists for all source words.

Compared to co-occurrence counts, this strategy avoids selecting frequent target words when conditioning on a rare source word. Word alignments will only link a frequent target word to a rare source word if no better explanation is present in the source sentence.

## 2.4 PHRASE PAIRS

This strategy relies on a phrase translation table, i.e., a table pairing source phrases with corresponding target phrases. The phrase table is constructed by reading off all bilingual phrases that are consistent with the word alignments according to an extraction heuristic (Koehn, 2010). For selection, we restrict the phrase table to the phrases present in the source sentence and consider the union of the word types appearing in all corresponding target phrases (Mi et al., 2016). Compared to word alignments, we hope this strategy to fetch more relevant target words as it can rely on longer source phrases to leverage richer source context.

## 2.5 SUPPORT VECTOR MACHINES

Support Vector Machines (SVMs) for vocabulary selection have been previously proposed in (Bangalore et al., 2007). The idea is to determine a target vocabulary based on the entire source sentence rather than individual words or phrases. In particular, we train one SVM per target word taking as input a sparse vector encoding the source sentence as a bag of words. The SVM then predicts whether the considered target word is present or absent from the target sentence.

This classifier-based method has several advantages compared to phrase alignments: the input is not restricted to a few contiguous source words and can leverage all words in the source sentence. The model can express anti-correlation with negative weights, marking that the presence of a source word is a negative indicator for the presence of a target word. A disadvantage of this approach is that we need to feed the source sentence to all SVMs in order to get scores, instead of just reading from a pre-computed table. However, SVMs can be evaluated efficiently since (i) the features are sparse, and (ii) only features corresponding to words from the source sentence are used at each evaluation. Finally, this framework formulates the selection of each target word as an independent binary classification problem which might not favor competition between target words.

## 2.6 COMMON WORDS

Following Mi et al. (2016), we consider adding the $k$ most frequent target words to the above selection methods. This set includes conjunctions, determiners, prepositions and frequent verbs. Pruning any such word through restrictive vocabulary selection may adversely affect the system output and is addressed by this technique.

## 3 RELATED WORK

The selection of a limited target vocabulary from the source sentence is classical topic in machine translation. It is often referred to as *lexical selection*. As mentioned above, word-based and phrase-based systems perform implicit lexical selection by building a word or phrase table from alignments to constrain the possible target words. Other approaches to lexical selection include discriminative models such as SVMs and Maximum Entropy models (Bangalore et al., 2007) as well as rule-based systems (Tufis, 2002; Tyers et al., 2012).

In the context of neural machine translation, vocabulary size has always been a concern. Various strategies have been proposed to improve training and decoding efficiency. Approaches inspired by importance sampling reduce the vocabulary for training (Jean et al., 2015a), byte pair encoding segment words into more frequent sub-units (Sennrich et al., 2016a), while (Luong & Manning, 2016) proposes to segment words into characters. Related work in neural language modeling is also relevant (Bengio et al., 2003; Mnih & Hinton, 2008; Chen et al., 2016). One can refer to (Sennrich, 2016) for further references.

Closer to our work, recent work (Mi et al., 2016) presents preliminary results on using lexical selection techniques in an NMT system. Compared to this work, we investigate more selection methods

(SVM, PCA, co-occurrences) and analyze the speed/accuracy trade-offs at various operating points. We report efficiency gains and distinguish the impact of selection in training and decoding.

# 4    EXPERIMENTS & RESULTS

This section presents our experimental setup, then discusses the impact of vocabulary selection at decoding time and then during training time.

## 4.1    EXPERIMENTAL SETUP

We use a an encoder-decoder style neural machine translation system based on Torch.[1] Our encoder is a bidirectional recurrent neural network (Long Short Term Memory, LSTM) and an LSTM decoder with attention. The resulting context vector is fed to an LSTM decoder which generates the output (Bahdanau et al., 2015; Luong et al., 2015a). We use a single layer setup both in the encoder and as well as the decoder, each with $512$ hidden units. Decoding experiments are run on a CPU since this is the most common type of hardware for inference. For training we use GPUs which are the most common hardware for neural network fitting. Specifically, we rely on 2.5GHz Intel Xeon 5 CPUs and Nvidia Tesla M40 GPUs. Decoding times are based on a single CPU core and training times are based on a single GPU card.

Word alignments are computed with FastAlign (Dyer et al., 2013) in both language directions and then symmetrized with 'grow-diag-final-and'. Phrase tables are computed with Moses (Koehn et al., 2007) and we train support vector machines with SvmSgd (Bottou, 2010). We also use the Moses preprocessing scripts to tokenize the training data.

We experiment on two language pairs. The majority of experiments are on WMT-15 English to German data (Bojar et al., 2015); we use newstest2013 for validation and newstest2010-2012 as well as newstest2014,2015 to present final test results. Training is restricted to sentences of no more than $50$ words which results in 3.6m sentence pairs. We chose the vocabulary sizes following the same methodology. We use the 100k most frequent words both for the source and target vocabulary. At decoding time we use a standard beam search with a beam width of 5 in all experiments. Unknown output words are simply replaced with the source word whose attention score is largest (Luong et al., 2015b).

We also experiment with WMT-16 English to Romanian data using a similar setting but allowing sentences of up to $125$ words (Bojar et al., 2016). Since the training set provided by WMT is limited to 600k sentence pairs, we add the synthetic training data provided by Sennrich et al. (2016b). This results in a total of 2.4m sentence pairs. Our source vocabulary comprises the 200k most frequent words and the target vocabulary contains 50k words.

## 4.2    SELECTION FOR EFFICIENT DECODING

Decoding efficiency of neural machine translation is still much lower than for traditional phrase-based translation. For NMT, the running time of beam search on a CPU is dominated by the last linear layer that computes a score for each target word. Vocabulary selection can therefore have a large impact on decoding speed. Figure 1 (left) shows that a reduced vocabulary of $\sim 460$ types (144 msec) can achieve a 10X speedup over using the full 100k-vocabulary ($\sim 1,600$ msec).

Next we investigate the impact of reduced vocabularies on accuracy. Figure 1 (right) compares BLEU for the various selection strategies on a wide range of vocabulary sizes. Broadly, there are two groups of techniques: first, co-occurrence counts and bilingual embeddings (PCA) are not able to match the baseline performance (Full 100k) even with over 5k candidate words per sentence. Second, even with fewer than $1,000$ candidates per sentence, word alignments, phrase pairs and SVMs nearly match the full vocabulary accuracy.

---

[1]We will release the code with the camera ready.

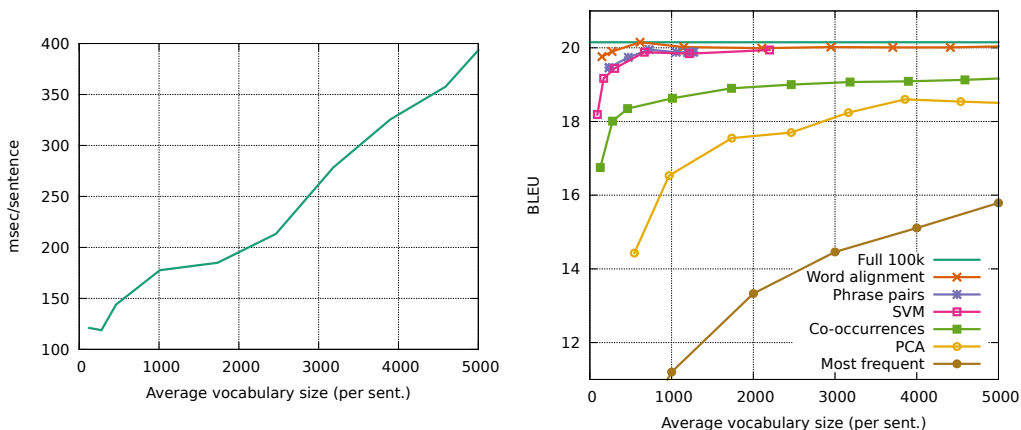

Figure 1: *Left:* Decoding time vs. vocabulary size on newstest2013 for WMT15 English to German translation. *Right:* BLEU vs. vocabulary size for different selection strategies.

Although co-occurrence counts and PCA have shown useful to measure semantic relatedness (Brown et al., 1992; Lebret & Collobert, 2014), it seems that considering the whole source sentence as the explanation of a target word without latent alignment variables undermines their selection ability. Overall, word alignments work as well or better than the other techniques relying on a wider input context (phrase pairs and SVMs). Querying a word table is also more efficient than querying a phrase-table or evaluating SVMs. We therefore use word alignment-based selection for the rest of our analysis.

Mi et al. (2016) suggest that adding common words to a selection technique could have a positive impact. We therefore consider adding the most frequent $k$ words to our word alignment-based selection. Figure 2 shows that this actually has little impact on BLEU in our setting. In fact, the overlap of the results for $n = 0$ and $50$ indicates that most of the top 50 words are already selected, even with small candidate sets.

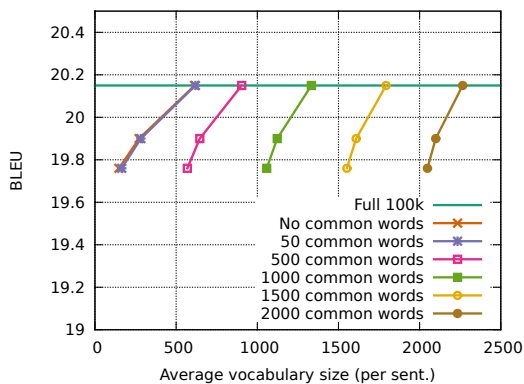

Figure 2: Impact of adding common words to word alignment selection for various vocabulary sizes.

Next we try to get a better sense of how precise selection is with respect to the words used by a human translator or with respect to the translations generated by the full vocabulary model. We use word alignments for this experiment. Figure 3 shows coverage with respect to the reference (left) and with respect to the output of the full vocabulary system (right). We do not count unknown words (UNK) in all settings, even if they may later be replaced by a source word (§4.1). Not counting UNKs is the reason why the full vocabulary models do not achieve 100% coverage in either setting. The two graphs show different trends: On the left, coverage with respect to the reference for the full vocabulary is 95.1%, while selection achieves 87.5% with a vocabulary of 614 words (3rd point on graph). However, when coverage is measured with respect to the full vocabulary system output,

the coverage of selection is very close to the full vocabulary model with respect to itself, i.e., when unknown words are not counted. In fact, the selection model covers over 99% of the non-UNK words in the full vocabulary output. This result shows that selection can recover almost all of the words which are effectively selected by a full vocabulary model while discarding many words which are not chosen by the full model.

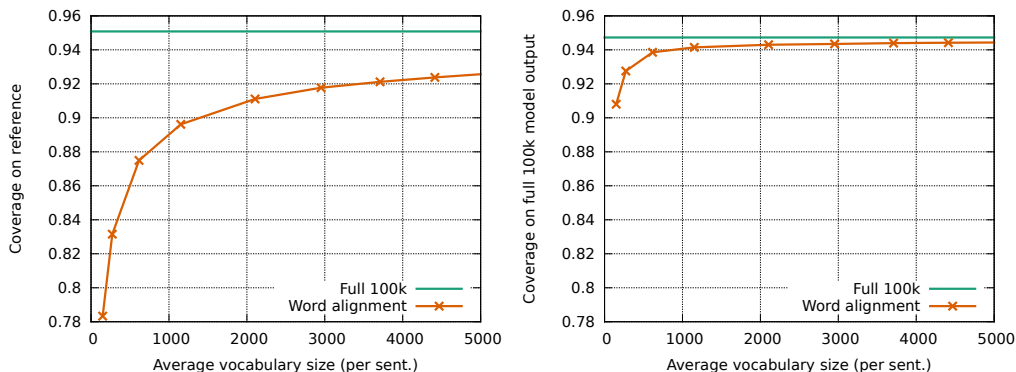

Figure 3: *Left:* Coverage of the reference by the word alignment selection for different vocabulary sizes and by the full 100k model. *Right:* Coverage of the full vocabulary model prediction by the word alignment selection method. Coverage does not count unknown words, therefore the full model has non-perfect coverage on itself. Vocabulary selection never fully covers the reference (left) but it almost entirely covers the prediction of the full vocabulary model, even when very few candidates are selected.

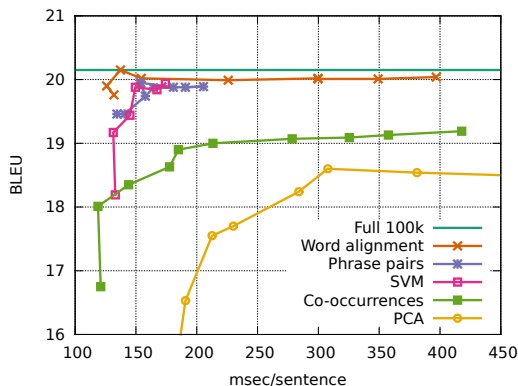

Figure 4: BLEU accuracy versus decoding speed for a beam size of 5 on CPU. Significant speed ups can be achieved with no decrease in BLEU accuracy, e.g., word alignment selection achieves 20.2 BLEU at 137 msec/sentence (156 words per sec) while the full vocabulary model requires 1,581 msec/sentence (13.5 words per sec) at the same accuracy level, this is equivalent to an 11-fold speed up.

What is the exact speed and accuracy trade-off when reducing the output vocabulary? Figure 4 plots BLEU against decoding speed. We pick a number of operating points from this graph for our final test set experiments (Table 1). For our best methods (word alignments, phrase alignments and SVMs) we pick points such that vocabularies are kept small while maintaining good accuracy compared to the full vocabulary setting. For co-occurrence counts and bilingual PCA we choose settings with comparable speed.

Our test results (Table 1) confirm the validation experiments. On English-German translation we achieve more than a 10-fold speed-up over the full vocabulary setting. Accuracy for the word alignment-based selection matches the full vocabulary setting on most test sets or decreases only slightly. For example with word alignment selection, the largest drop is on newstest2015 which

| EN-DE | Max. | 2010 | 2011 | 2012 | 2014 | 2015 | Voc. | Cov. | Time | Speed |
|---|---|---|---|---|---|---|---|---|---|---|
| Full vocab | – | 18.5 | 16.5 | 16.8 | 19.0 | 22.5 | 100,000 | 93.3% | 1,524 | 13 |
| Co-occur. | 300 | 17.2 | 15.6 | 15.8 | 18.1 | 20.6 | 1,036 | 81.1% | 156 | 141 |
| PCA | 100 | 15.4 | 13.7 | 14.2 | 14.5 | 18.6 | 966 | 74.8% | 143 | 144 |
| Word align | 100 | 18.5 | 16.4 | 16.7 | 19.0 | 22.2 | 1,093 | 88.5% | 143 | 144 |
| Phrases | 200 | 18.1 | 16.2 | 16.6 | 18.9 | 22.0 | 857 | 86.2% | 153 | 135 |
| SVM | – | 18.3 | 16.2 | 16.6 | 18.8 | 21.9 | 1,284 | 86.6% | – | – |

| EN-RO | Max. | | | | | 2016 | Voc. | Cov. | Time | Speed |
|---|---|---|---|---|---|---|---|---|---|---|
| Full vocab | – | | | | | 27.9 | 50,000 | 96.0% | 966 | 26 |
| Word align | 50 | | | | | 28.1 | 691 | 89.3% | 186 | 136 |

Table 1: Final decoding accuracy results for WMT English-German and English-Romanian on various test sets (newstest 2010 – 2016, except 2013 our validation set). We report the average vocabulary size per sentence, coverage of the reference and decoding time in milliseconds per sentence for newstest2015 and newstest2016. Decoding speed is reported in words per second. All timings are measured on the same machine using a single CPU core. The Max. column indicates the maximum number of selected candidates per source word or phrase.

achieves 22.2 BLEU compared to 22.5 BLEU for the full setting on English-German; the best single-system neural setup at WMT15 achieved 22.4 BLEU on this dataset (Jean et al., 2015b). On English-Romanian, we achieve a speed-up of over 5 times with word alignments at 28.1 BLEU versus 27.9 BLEU for the full vocabulary baseline. This matches the state-of-the-art on this dataset (Sennrich et al., 2016b) from WTM16. The smaller speed-up on English-Romanian is due to the smaller vocab of the baseline in this setting which is 50k compared to 100k for English-German.

## 4.3 SELECTION FOR BETTER TRAINING

So far our evaluation focused on vocabulary selection for decoding, relying on a model trained with the full vocabulary. Next we address the question of whether the efficiency advantages observed for decoding translate to training as well. Selection at training may impact generalization performance either way: it assimilates training and testing conditions which could positively impact accuracy. However, training with a reduced vocabulary could result in worse parameter estimates, especially for rare words which would receive much fewer updates because they would be selected less often.

We run training experiments on WMT English to German with word alignment-based selection. In addition to the selected words, we include the target words of the reference and train a batch with the union of the sentence-specific vocabularies of all samples (Mi et al., 2016).

Figure 5 compares validation accuracy of models trained with selection or with the full vocabulary. Selection in both training and decoding gives a small accuracy improvement. However, this improvements disappears for vocabulary sizes of 500 and larger; we found the same pattern on other test sets. Similar to our decoding experiments, adding common words during training did not improve accuracy. Table 2 shows the impact of selection on training speed. Our bi-directional LSTM model (BLSTM) can process the same number of samples in 25% less time on a GPU with a batch size of 32 sentences. We do not observe changes in the number of epochs required to obtain the best validation BLEU.

The speed-ups for training are significantly smaller than for decoding (Table 1). This is because training scores the vocabulary exactly once per target position, while beam search has to score multiple hypotheses at each generation step.

We suspect that training is now dominated by the bi-directional LSTM encoder. To confirm this, we replaced the encoder with a simple average pooling model which encodes source words as the mean of word and position embeddings over a local context (Ranzato et al., 2016). Table 2 shows

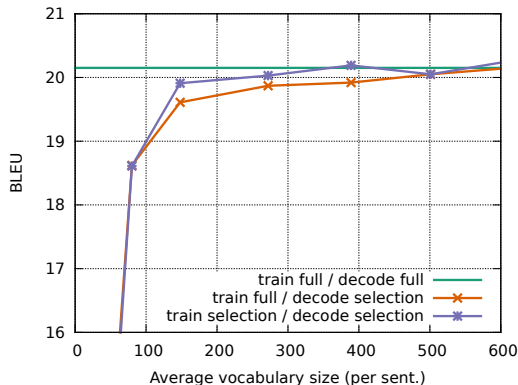

Figure 5: Accuracy on the validation set for different vocabulary sizes when using word alignment-based selection during training and testing, or the full vocabulary.

that in this setting the efficiency gains of vocabulary selection are more substantial (40% less time per epoch). This model is not as accurate and achieves only 18.5 BLEU on newstest2015 compared to 22.5 for the bi-directional LSTM encoder. However, it shows that improving the efficiency of the encoder is a promising future work direction.

| Vocab. per batch | 100k | 6k | |
|---|---|---|---|
| Avg. pooling encoder | 5h 55 | 3h 34 | (-40%) |
| BLSTM encoder | 9h 34 | 7h 13 | (-25%) |

Table 2: Training times per epoch over 3.6m sentences in hours and minutes on German-English for the full (100k) and reduced vocabulary settings (6k). Measurements include forward/backward/update on a GPU for a batch of size 32. The 6k candidate words per batch correspond to an average of 390 words per sentence.

## 5 CONCLUSIONS

This paper presents a comprehensive analysis of vocabulary selection techniques for neural machine translation. Vocabulary selection constrains the output words to be scored to a small subset relevant to the current source sentence. The idea is to avoid scoring a high number of unlikely candidates with the full model which can be ruled out by simpler means.

We extend previous work by considering a wide range of simple and complex selection techniques including bilingual word co-occurrence counts, bilingual embeddings built with Hellinger PCA, word alignments, phrase pairs, and discriminative SVM classifiers. We explore the trade-off between speed and accuracy for different vocabulary sizes and validate results on two language pairs and several test sets.

Our experiments show that decoding speed-up can be reduced by up to 90% without compromising accuracy. Word alignments, bilingual phrases and SVMs can achieve high accuracy, even when considering fewer than 1,000 word types per sentence.

At training time, we achieve a speed-up of up to 1.33 with a bi-directional LSTM encoder and 1.66 with a faster alternative. Efficiency increases are less pronounced during training because of two combined factors. First, vocabulary scoring at the final layer of the model is a smaller part of the computation compared to beam search. Second, state-of-the-art bi-directional LSTM encoders (Bahdanau et al., 2015) are relatively costly compared to scoring the vocabulary on GPU hardware. Efficiency gains from vocabulary selection highlight the importance of progress towards efficient, accurate encoder and decoder architectures.

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
