# Peer review of "Vocabulary Selection Strategies for Neural Machine Translation"

_ICLR 2017 — rejected_

[Official Review · AnonReviewer4 · rating 5 · confidence 3 · 16 Dec 2016]
**Useful tricks for faster decoding and training of NMT**

This paper evaluates several strategies to reduce output vocabulary size in order to speed up NMT decoding and training. It could be quite useful to practitioners, although the main contributions of the paper seem somewhat orthogonal to representation learning and neural networks, and I am not sure ICLR is the ideal venue for this work.

- Do the reported decoding times take into account the vocabulary reduction step?
- Aside from machine translation, might there be applications to other settings such as language modeling, where large vocabulary is also a scalability challenge?
- The proposed methods are helpful because of the difficulties induced by using a word-level model. But (at least in my opinion) starting from a character or even lower-level abstraction seems to be the obvious solution to the huge vocabulary problem.

[Official Review · AnonReviewer2 · rating 4 · confidence 4 · 17 Dec 2016]
**A well-executed NLP paper but with little novelty**

In this paper, the authors present several strategies to select a small subset of target vocabulary to work with per source sentence, which results in significant speedup. The results are convincing and I think this paper offers practical values to general seq2seq approaches to language tasks. However, there is little novelty in this work: the authors further mostly extend the work of (Mi et al., 2016) with more vocabulary selection strategies and thorough experiments. This paper will fit better in an NLP venue.

[Official Review · AnonReviewer5 · rating 4 · confidence 5 · 30 Dec 2016]
**vocabulary selection is a very promising technique**

This paper compares several strategies for guessing a short list of vocabulary for the target language in neural machine translation. The primary findings are that word alignment dictionaries work better than a variety of other techniques.

My take on this paper is that to have a significant impact, it needs to make the case for why one might want vocabulary rather than characters or sub word units like BPE. I think there are likely many very good reasons to do this that could be argued for (synthesize morphology, deal with transliteration, etc), but most of these would suggest some particular models and experiments, which are of course not in this paper. As it is, I think this paper is a useful but minor contribution that shows that word alignment is a good way of getting short lists, but it does not strongly make the case that we should abandon work in other directions.

Minor comments:
In addition to the SVM approach for modeling vocabulary, the discriminative word lexicon of Mauser et al. (2009) and the neural version of Ha et al. (2014) are also worth mentioning.

It would be useful to know what the coverage rate of the actual full vocabulary would be (rather than the 100k “full vocabulary”). Since presumably this technique could be used to work with much larger vocabularies.

When reducing the vocabulary size for training, the Mi et al. (2016) technique of taking the union of all the vocabularies in a mini batch seems like a rather strange objective. If the vocabulary of a single sentence is used, the probabilistic semantics of the translation model can still be preserved since p(e | f, vocab(f)) = p(e | f) if p(vocab(f) | f) = 1, i.e., is deterministic, which it is here. Whereas the objective is no longer a sensible probability model in the mini batch vocabulary case. Thus, while it may be a bit more difficult to implement, it seems like it would at least be a sensible comparison to make.

[Official Review · AnonReviewer3 · rating 5 · confidence 3 · 05 Jan 2017]
**A solid, practical paper - but not very innovative**

This paper conducts a comprehensive series of experiments on vocabulary selection strategies to reduce the computational cost of neural machine translation.

A range of techniques are investigated, ranging from very simple methods such as word co-occurences, to the relatively complex use of SVMs.

The experiments are solid, comprehensive and very useful in practical terms.  It is good to see that the best vocabulary selection method is very effective at achieving a very high proportion of the coverage of the full-vocabulary model (fig 3).  However, I feel that the experiments in section 4.3 (vocabulary selection during training) was rather limited in their scope - I would have liked to see more experiments here.

A major criticism I have with this paper is that there is little novelty here.  The techniques are mostly standard methods and rather simple, and in particular, there it seems that there is not much additional material beyond the work of Mi et al (2016).  So although the work is solid, the lack of originality lets it down.

Minor comments: in 2.1, the word co-occurence measure - was any smoothing used to make this measure more robust to low counts?

[Final Decision · Program Chairs · 06 Feb 2017]
**ICLR committee final decision**

The reviewers agree that the method is exciting as practical contributions go, but the case for originality is not strong enough.